# Maternal Neutrophil Depletion Fails to Avert Systemic Lipopolysaccharide-Induced Early Pregnancy Defects in Mice

**DOI:** 10.3390/ijms22157932

**Published:** 2021-07-25

**Authors:** Sourav Panja, John T. Benjamin, Bibhash C. Paria

**Affiliations:** Division of Neonatology, Department of Pediatrics, Vanderbilt University Medical Center, Nashville, TN 37232, USA; sourav.panja@vumc.org (S.P.); john.benjamin@vumc.org (J.T.B.)

**Keywords:** leukocytes, chemokines, cytokines, lipopolysaccharide, early pregnancy, uterus, implantation, decidual stromal zone, non-decidual stromal zone

## Abstract

Maternal infection-induced early pregnancy complications arise from perturbation of the immune environment at the uterine early blastocyst implantation site (EBIS), yet the underlying mechanisms remain unclear. Here, we demonstrated in a mouse model that the progression of normal pregnancy from days 4 to 6 induced steady migration of leukocytes away from the uterine decidual stromal zone (DSZ) that surrounds the implanted blastocyst. Uterine macrophages were found to be CD206^+^ M2-polarized. While monocytes were nearly absent in the DSZ, DSZ cells were found to express monocyte marker protein Ly6C. Systemic endotoxic lipopolysaccharide (LPS) exposure on day 5 of pregnancy led to: (1) rapid (at 2 h) induction of neutrophil chemoattractants that promoted huge neutrophil infiltrations at the EBISs by 24 h; (2) rapid (at 2 h) elevation of mRNA levels of MyD88, but not Trif, modulated cytokines at the EBISs; and (3) dose-dependent EBIS defects by day 7 of pregnancy. Yet, elimination of maternal neutrophils using anti-Ly6G antibody prior to LPS exposure failed to avert LPS-induced EBIS defects allowing us to suggest that activation of Tlr4-MyD88 dependent inflammatory pathway is involved in LPS-induced defects at EBISs. Thus, blocking the activation of the Tlr4-MyD88 signaling pathway may be an interesting approach to prevent infection-induced pathology at EBISs.

## 1. Introduction

Early pregnancy loss is a common occurrence in humans. While about 10–25% of all clinically recognized pregnancies result in failure, a significant number of pregnancies even fail prior to being recognized [1]. Untreated infection during prepregnancy as well as new infection after conception in reproductive and non-reproductive tissues remains a threat for pregnancy loss and/or complications [2,3,4,5]. Although antibiotic therapy is prudent for saving pregnancy from infection, its use remains worrisome given its adverse effects on pregnancy outcome and fetal growth, concern for alteration of the maternal and fetal microbiome, and the emergence of antibiotic-resistant bacteria. Therefore, the development of new therapies alternative to antibiotics for preventing infection-induced inflammatory changes at the early blastocyst implantation site (EBIS) is needed. This can be achieved if molecular determinants of LPS-induced harmful inflammation at the EBIS are identified by studying the full spectrum of physiological as well as infection-induced immune cell and inflammatory changes at the EBIS of the uterus.

The uterine role to carry a pregnancy to term starts from the moment when a blastocyst implants on the uterine endometrium. Blastocyst implantation is a natural event, but it is also considered a perilous incident due to the semi-allogenic as well as pro-inflammatory nature of the blastocyst [6]. It is acknowledged that the initiation of blastocyst implantation induces a prompt local inflammatory reaction at its uterine attachment site [7,8]. Compared with the traditional microbial infection-induced immune-inflammatory reaction that is developed to destroy pathogens [9], the acute blastocyst-induced inflammation at the uterine attachment site is unique because it is harmless to the implanted semi-allogenic blastocyst. Until now, it remains largely unclear how blastocyst-induced uterine inflammation at the EBIS remains in check without harming the blastocyst.

Leukocytes are the main building blocks of the innate immune system and mount an inflammatory response to counteract foreign entities. Although distinct subpopulations of leukocytes including macrophages (MΦs), dendritic cells (DCs), natural killer (NK) cells, and neutrophils (NPs) exist and participate in the host immune response, NPs are the most prevalent [10]. NPs migrate to sites of inflammation to target inflammation-causing agents [11]. Chemokines including C-X-C motif chemokine ligand 1 (Cxcl1) and C-X-C motif chemokine ligand 2 (Cxcl2) are considered as strong chemoattractants for NPs [12]. Given the inflammatory nature of the blastocyst [13] as well as the blastocyst–uterine attachment reaction [8], migration of maternal leukocytes towards the implanted blastocyst is a possibility and may cause tissue damage. However, immune events and inflammation that occur at the normal BIS must be implantation friendly to allow a successful pregnancy. Thus, the migration of leukocytes away from the core EBIS could be a biologically relevant mechanism of generating an immune-privileged core EBIS. There is evidence that MΦs escape from the decidual stromal zone (DSZ) to the non-decidualized stromal zone (NDSZ) suggesting distancing of these cells from the decidual event of implantation [14,15]. On the contrary, it has previously been demonstrated that DCs remain trapped within the decidua to promote immunological acceptance of the blastocyst [16,17]. Most immune cell studies at the EBIS suggest that the decidua creates a milieu that restricts infiltration of specific leukocytes to promote immunological acceptance of the fetus. Moreover, studies have shown that certain types of leukocytes such as DCs, NK cells and others are contributors to events associated with implantation such as decidualization, trophoblast invasion and decidual angiogenesis irrespective of their location at the EBIS [18,19,20,21,22,23,24]. Thus, when uterine leukocyte populations at the EBIS are dysregulated in number and/or function due to maternal infection, the ability of the uterus to protect the embryo as well the pregnancy is likely to be disturbed.

Endotoxic lipopolysaccharide (LPS), a component of the outer membrane of Gram-negative bacteria, is a potent inflammogen and is frequently used to discern mechanisms of infection-induced pregnancy complications at various stages of pregnancy in the mouse model. Among Toll-like receptors (Tlrs), Tlr4 acts as the primary signaling receptor for LPS [25]. A receptor complex involving Tlr4 and cluster of differentiation 14 (Cd14) after binding LPS recruits either adaptor protein myeloid differentiation factor 88 (MyD88) or toll-interleukine-1 receptor domain-containing adaptor-inducing interferon-β (Trif) for signaling [26]. While the MyD88-dependent Tlr4 signaling pathway is accountable for producing proinflammatory cytokines such as tumor necrosis factor-alpha (Tnfα), interleukin 1β (Il1β) and interleukin 6 (Il6), the Trif-dependent Tlr4 signaling pathway triggers interferon response factors to produce type-1 interferons (Ifns), Ifnα and Ifnβ [27]. The role of these two different Tlr4-mediated downstream pathways at the EBIS is largely unknown. It is possible that the activation of these pathways may be event-specific. Tlr4 is expressed in both decidual [28] and circulating innate immune cells [27]. Hence, LPS could be signaling via circulatory immune cells and uterine decidual cells or both.

In the current study, immunofluorescent tools were applied to demonstrate leukocyte landscape at the normal as well as LPS-exposed EBISs. We studied abundance and localization sites of leukocyte subtypes in normal EBISs to reveal the specific leukocyte population that is recruited at the mouse EBIS in response to a dose of LPS that is enough to cause early pregnancy defects. Our study provides a few important observations. First, the event of blastocyst implantation onto the uterine endometrium causes leukocyte fugetaxis from the endometrial DSZ to the NDSZ. Second, maternal exposure to LPS induces overwhelming migration of NPs to the NDSZ of the EBIS. Third, our results uncover that depletion of maternal circulating NPs effectively blocked LPS-induced migration of NPs to the EBIS, but was ineffective to avert LPS-induced defects at the EBIS. Lastly, by investigating the profile of mRNA expression of MyD88, Trif, and four cytokines including Tnfα, Il1β, Il6 and Ifnβ at the EBIS following LPS exposure, our study suggests that LPS causes rapid induction of Tlr4-MyD88, but not Tlr4-Trif, signaling pathway at the EBIS.

## 2. Results

### 2.1. Spatio-Temporal Distribution of Leukocytes in the Day 4 Receptive Uterus and Days 5 and 6 EBISs

We examined the preponderance and distribution of total leukocytes and their subtypes on uterine sections obtained every 12 h interval during the peri-implantation period starting from day 4 receptive uterus to day 6 of pregnancy to determine whether initiation of blastocyst implantation and a day of the natural progression of pregnancy have any major impact on leukocyte distribution and the population at the EBIS. In mice, the beginning of blastocyst implantation is detected as early as the late night of day 4 or as late as the early morning of day 5 [29]. It is generally considered that the bicornuate mouse uterus attains receptivity for blastocyst implantation on day 4 of pregnancy after a small elevation of the circulatory estradiol-17β level at around the noon time of this day [30]. Thus, we considered that uterine horns are adequately prepared on the evening of day 4 for initiating blastocyst–uterus communication and the process of implantation.

Total Leukocyte Distribution in the Day 4 Receptive Uterus and Days 5 and 6 EBISs (Figure 1.)

The cluster of differentiation 45 (CD45) antigen, also known as leukocyte common antigen or protein tyrosine phosphatase, receptor type-C, is expressed on all leukocytes [31]. Immunofluorescent imaging revealed that CD45^+^ immune cells were in both endometrial and myometrial uterine tissue compartments in the day 4 receptive uterus and were evenly distributed within two compartments. CD45^+^ cells were more abundant in the myometrium that is composed of an inner circular layer and an outer longitudinal layer compared with the endometrium of the day 5 (09 h) EBIS. Cells positive for CD45 were evenly distributed within the endometrium. As pregnancy and decidualization progressed from day 5 to day 6, a gradual reduction of CD45^+^ cells was observed in the DSZ surrounding the implanted blastocyst with an accumulation of these cells at the endometrial NDSZ in between the circular muscle layer and DSZ (Figure 1).

Prior studies indicate that lymphocyte populations within the uterine endometrium are very low [32] and uterine NK (uNK) cells do not appear before day 7 of pregnancy [33]. Thus, we specifically examined the abundance and distribution of NPs and monocytes, MΦs and DCs, in the normal day 4 receptive uterus and at the EBIS.

Identification of lymphocyte antigen 6 complex locus G6D (Ly6G) positive neutrophils and lymphocyte antigen 6 complex locus C (Ly6C) positive monocytes in the day 4 receptive uterus and days 5 and 6 EBISs.

Ly6G is a specific marker that separates neutrophils from other leukocytes [34]. On day 4 evening, Ly6G^+^ immune cells were barely detected in the myometrium and a few are detected within the endometrium. We also found only a few Ly6G^+^ immune cells both in the endometrium and myometrium on day 5 BIS. However, a clear presence of a few Ly6G^+^ immune cells was also observed in the subepithelial stroma close to the implantation chamber. On day 5 evening and day 6 morning, a few Ly6G^+^ immune cells were primarily found in the myometrium and NDSZ while they were hardly detected within the DSZ (Figure 2).

Ly6C is expressed in ~90% of the monocytes produced by the bone marrow [35]. We did not observe any Ly6C^+^ immune cells within any uterine cell compartment of day 4 receptive uterus and days 5 and 6 EBISs. However, we observed that uterine cells nearer to the implanted blastocyst were Ly6C^+^. Since CD45 staining was negative in these cells, we presumed that decidual stromal cells surrounding the implanted embryo on day 6 of pregnancy were Ly6C^+^ (Figure 2).

Having seen Ly6C^+^ staining in decidual cells we next applied RNAscope to detect and visualize Ly6C mRNA expression in these cells (Figure 3). We observed discrete puncta of signals of both Ly6C protein (Figure 3a) and mRNA (Figure 3b) in the similar site of the decidua surrounding the implanted embryo indicating the ability of these cells to synthesize Ly6C protein. While sections stained with mouse tissue non-specific alkaline phosphatase gene (*Alpl)*, a known marker of uterine decidual cells [28], produced positive signals in decidual cells, sections stained with dihydrodipicolinate reductase *(Dapb)* gene probe of *Bacillus subtilis* did not exhibit any positive signals (Figure 3b).

Distribution patterns of MΦs and DCs in the day 4 receptive uterus and days 5 and 6 EBISs.

In mice, there are two monocyte subtypes, Ly6C^+^ and Ly6C^−^ [35]. Based on our results in the preceding sections (Figure 2), we failed to identify leukocytes positive for Ly6C in the peri-implantation uterus (days 4-6 of pregnancy). Thus, we proceeded to identify MΦs and DCs. MΦs were identified by their expressions of a cell surface glycoprotein F4/80 [36] and CD206 [37]. CD206 is not expressed in M1 MΦs [37] and therefore, F4/80^+^, CD206^−^ and F4/80^+^, CD206^+^ are considered as M1 MΦs and M2 MΦs, respectively. CD11c is a specific marker of DCs [38].

1. Distribution of MΦs and DCs 

On the evening of day 4 and the morning of day 5, F4/80^+^ MΦs were abundant on the myometrium and endometrium of the uterus and evenly distributed. As pregnancy progresses to day 6, F4/80^+^ MΦs were scattered within DSZ and NDSZ of the EBIS. While a small number of F4/80^+^ cells were located inside the DSZ, most F4/80^+^ MΦs were observed at the NDSZ (Figure 4). However, the abundance of these was much more in the myometrium as compared with the endometrium (Figure 4). The abundance and distribution patterns of CD206^+^ cells were like F4/80^+^ cells (Figure 4). The distribution of CD11c^+^ cells was even throughout the myometrium and endometrium of the uterus on the evening of day 4. While the distribution of endometrial CD11c^+^ cells remains unaltered on day 5 EBIS, their abundance was more in the myometrium as compared to day 4 uterus. CD11c^+^ positive cells were mainly found in the NDSZ of the endometrium and myometrium at the EBIS of day 5 evening and day 6 morning. These cells were scarce within the DSZ (Figure 4).

2. The phenotypic differences of MΦs (M1 and M2 MΦs) 

Uterine sections when double immunostained with F4/80 and CD206 (Figure 5a) exhibited that most of the myometrial and endometrial F4/80^+^ cells were also CD206^+^ in all four time points of early pregnancy (Figure 5a). These findings suggested that the M2 MΦs subtype is predominant in the day 4 receptive uterus and days 5 and 6 EBISs during the peri-implantation period of pregnancy.

3. CD11c^+^ immune cells within the NDSZ are not F4/80^+^ MΦs

Although CD11c is a traditional marker of DCs, several studies have also found it to be expressed by M1 state MΦs [39]. Co-localization of CD11c and F4/80 illustrated that while a subset of myometrial F4/80 MΦs showed reactivity for CD11c antigen, most endometrial F4/80^+^ macrophages were negative for CD11c (Figure 5b) at four time points of early pregnancy.

### 2.2. Maternal LPS-Exposure Dose-Dependently Causes Defects/Loss of EBISs

In initial experiments, pregnant mice exposed to 1 µg LPS on day 5 of pregnancy when euthanized on day 8 pregnancy exhibited total implantation failure with no sign of EBISs. Therefore, a decision was made to euthanize mice on day 7 of pregnancy following LPS injections. Injection of vehicle (saline) and monophosphoryl lipid A (MPLA), a detoxified LPS-variant [40], to pregnant mice on day 5 of pregnancy is inefficacious at exhibiting any pregnancy faults on day 7 given the presence of visible normal-looking EBISs (Figure 6a) and a comparable number of EBISs (Figure 6b) with equivalent wet weight (Figure 6c) as ordinarily seen in untreated pregnant mice. All concentrations of LPS tested did not have any effect on the number of EBISs at day 7 of pregnancy (Figure 6b). Injecting pregnant mice with a lower dose of LPS (0.1 µg/mice) showed no reduction in the wet weight of EBISs compared to vehicles. However, pregnant mice receiving higher LPS doses (0.50, 0.75 and 1.0 µg) showed a dose-dependent but significant reduction (*p* < 0.05) in the wet weight of each EBIS compared to vehicle, MPLA and a lower dose (0.1 µg/mice) of LPS groups (Figure 6c). A gross histological examination of day 7 EBISs revealed gradual deterioration of EBISs with worsening decidual structure following 0.75 and 1.0 µg of LPS exposure compared with vehicle, MPLA and 0.1 µg of LPS exposure (Figure 6a). Mice receiving 1.0 µg LPS revealed very small EBISs with the frequent absence of the implanted embryo suggesting decay of the EBIS and demise of the embryo (Figure 6a).

#### 2.2.1. LPS-Induced Inflammatory Response at the Day 5 EBIS 

We predicted that LPS action would cause alterations in genes that are involved in leukocyte recruitment and inflammation at the day 5 EBIS. Compared with EBISs of the vehicle group, EBISs of the LPS-treated group showed significantly elevated levels of *Cxcl1*, *Cxcl2*, *MyD88*, *Tnfα*, *Il1β* and *Il6* mRNAs, but not *Trif* and *Ifnβ* mRNAs. LPS exposure for 2 h increased mRNA expression levels of *Cxcl1* and *Cxcl2* by 453- and 1476-fold, respectively (Figure 7a); *MyD88* by 4-fold (Figure 7b); and *Tnfα*, *Ilβ* and *Il6* by 9-, 40- and 426-fold, respectively (Figure 7c). The analysis of these above-described genes at the EBIS following the MPLA treatment demonstrated no significant differences in their expression levels as compared to the vehicle treatment.

#### 2.2.2. LPS-Mediated Neutrophil Accumulation at the EBIS

Because pregnant mice exposed to 1 µg of LPS on day 5 of pregnancy showed visible impaired EBISs on day 7 of pregnancy, we harvested day 6 EBISs from vehicle-, MPLA- and LPS-injected pregnant mice to determine the influence of LPS on the infiltration of specific leukocyte subtypes including NP and monocyte at the EBIS. The immunofluorescence microscopy results revealed that as compared to vehicle and MPLA treatments, LPS treatment induced Ly6G^+^ NP infiltration to the DSZ, NDSZ and myometrium of the EBIS (Figure 8a). The magnitude of fold change upon LPS exposure for NPs was significantly (*p* < 0.01) higher in the DSZ (~9-fold), NDSZ (~115-fold) and myometrium (~42-fold) than the vehicle. The largest fold-increase in neutrophilic infiltration was noted in the NDSZ in response to LPS. No differences in fold change were noted between the vehicle- and MPLA-treated groups (Figure 8b).

Visual analyses of the abundance of monocyte subtypes demonstrated that LPS exposure was unsuccessful to cause any obvious increase gathering of F4/80^+^ and CD206^+^ macrophages and CD11c^+^ cells in all three compartments of EBISs as compared to vehicle and MPLA treatments (Figure 8a). These cells were primarily localized in the myometrium and NDSZ (Figure 8a) of the EBIS. However, cell number counting and fold change analysis in three compartments of the EBIS demonstrated a small but significant (<0.05) decrease only in the myometrium in response to LPS exposure as compared to vehicle and MPLA exposures (Figure 8b).

#### 2.2.3. Administration of anti-Ly6G antibody Prevented LPS-Induced NP Accumulation at the EBIS

The specific antibody for neutrophil depletion in rodent models is anti-Ly6G [41]. Representative immunofluorescence images presented in Figure 9a revealed that the accumulation of ly6G^+^ NPs that was observed in the uterine myometrium and the NDSZ of the day 6 EBIS of LPS alone or LPS plus IgG2b isotype antibody-treated control mice did not occur in the EBIS of mice treated with anti-ly6G plus LPS. When we quantified and compared the number of ly6G^+^ NPs at EBISs of all treatment groups, we revealed almost complete removal of NPs from anti-ly6G plus LPS treated groups compared with LPS alone or LPS plus IgG2b isotype antibody-treated groups (Figure 9b). These findings indicated that pretreatment with anti-Ly6G prior to LPS exposure prevented LPS-induced circulating NP influx at the EBIS.

#### 2.2.4. NP Depletion Prior to LPS Treatment Fails to Prevent LPS-induced EBIS Defects

Because LPS treatment on day 5 of pregnancy induces NP infiltration at the EBIS by day 6 and visible EBIS defects by day 7, we determined whether NP migration at the EBIS is responsible for the induction of LPS-induced EBIS defects. We noted no reduction in the number of EBISs and wet weight of each EBIS in the anti-Ly6G group compared with the vehicle group (Table 1). While LPS exposure alone exhibited EBIS defects in 100% (13/13) of mice, mice treated with anti-ly6G plus LPS (5/5) failed to reverse any EBIS defects. The number of EBISs and wet weight of each EBIS in anti-ly6G plus LPS treated group remain almost like groups that received LPS and anti-IgG2b plus LPS (Table 1).

Pregnant mice were divided into five groups. Each mouse in groups 1 and 2 received a single injection (i.p.) of vehicle (0.1 mL of saline) and anti-Ly6G Ab (150 µg/0.1 mL saline), respectively, on day 4. Each mouse in group 3 received a single injection (i.p.) of LPS (1 µg/0.1 mL saline) on day 5. Each mouse in groups 4 and 5 received an injection (i.p.) of anti-IgG2b (150 µg/0.1 mL saline) and anti-Ly6G (150 µg/0.1 mL saline) antibodies, respectively, on day 4 of pregnancy prior to LPS injection (i.p.) on Day 5. Pregnancy outcomes in mice from all groups were analyzed on day 7 of pregnancy. The number (#) and wet weight of each implantation site were recorded. Differences in the wet weight of EBISs from groups 2–5 were compared with group 1. Results are presented as mean ± SEM. *** *p* <0.005, significantly different from group 1 (ANOVA followed by Tukey test).

## 3. Discussion

This investigation yielded several key findings: (1) the endometrial DSZ that surrounds the implanted blastocyst is virtually an immune cell-free zone; (2) Ly6C^+^ monocytes are not present in the peri-implantation uterus; (3) endometrial DSZ cells nearer to the implanted blastocyst are Ly6C^+^; (4) maternal LPS exposure after initiation of implantation induces expression of MyD88 and its downstream inflammatory cytokines genes, NP infiltration, and defects at the EBIS and (5) however, infiltrating NPs do not appear to play a key role in triggering LPS-induced defects at the EBIS.

We created a comprehensive picture of the innate immune cell populations in the peri-implantation uterus. This is critical from an immunological perspective because the semi-allogenic blastocyst requires a unique uterine immune milieu during the peri-implantation period to avoid attack by uterine immune cells. Leukocytes are innate immune responders with abilities to identifying, destroying, and removing pathogens, dead and muted cells, and foreign entities. The presence of evenly distributed total leukocytes (CD45^+^ cells) and their subtypes in the endometrium of the day 4 receptive uterus and day 5 implantation site indicates their role in fighting infections and eliminating dead cells or foreign entities. However, we observed a change in the topography of uterine immune cells after completion of the initial blastocyst–uterine attachment reaction. Day 5 late and day 6 EBISs showed only a few scattering leukocytes in the DSZ surrounding the embryo (the core EBIS) and preponderance of leukocytes in the NDSZ and myometrium (outside the core EBIS). These observations support the established view that the core EBIS is a “leukocyte restricted zone” [14,42]. It is likely that the unconventional attachment between the uterine luminal epithelial cells with the trophoblast epithelial cell of the blastocyst and subsequent stromal cell decidualization repels instead of attracting leukocytes within the core BIS. It is also possible that the microenvironment of the blastocyst-induced DSZ does not favor the survival of leukocytes, but the microenvironment of NDSZ does. Fibroblasts are diverse cell types that display topographic differentiation and positional memory. Thus, site-specific differences in fibroblast populations may account for the ability of different stromal microenvironments to induce the differential accumulation of leukocytes [43,44]. Previous studies have also shown that the decidua functions as an anatomical and permeability barrier zone [45]. These inherent decidual properties may help to keep the leukocytes away from it in order to hide the newly implanted blastocyst from the immune cells.

In our study, leukocyte subtype analysis at the normal EBIS revealed that the relative proportions of F4/80/CD206 monocyte/MΦ subpopulations in the NDSZ and myometrium were much more than Ly6G granulocytes and CD11c monocyte subtypes. As NPs are the body’s first defense against any foreign entities [46], the presence of only a few NPs at the normal EBIS may provide an evolutionary benefit to the survival of the semiallogenic blastocyst and maintenance of a threshold level of inflammation at this site. Murine monocytes are commonly divided into two subsets: Ly6C^+^ classical or inflammatory subtype and Ly6C^−^ non-classical or resident subtype [47,48]. Their distribution at the normal EBIS is not sufficiently defined. Our immunophenotyping of Ly6C demonstrated the absence of Ly6C^+^ immune cells in the day 4 uterus and days 5 and 6 EBISs suggesting uterine monocytes are Ly6C^−^. As monocytes differentiate into MΦs and DCs, we surmise that Ly6C^−^ monocytes upon entering inside the endometrium differentiated into residential MΦs and DCs. Tissue-resident MΦs and DCs are shown to play important role in the development and tissue homeostasis [49]. Mice depleted of CD11c MΦs or DCs exhibit implantation problems due to impaired uterine receptivity or decidualization and embryo implantation [50,51]. MΦs are phagocytic cells like NPs and acquire distinct polarization states from proinflammatory (M1) to anti-inflammatory (M2) depending on the tissue microenvironment [49]. While M1 MΦs secrete proinflammatory cytokines, M2 MΦs are anti-inflammatory and immunosuppressive cells [52]. In this study, we identified a predominance of M2 MΦs in the receptive uterus as well in the EBISs as demonstrated by their F4/80 and CD206 double-positive phenotype. These findings indicate that these M2 MΦs perhaps play an immune suppression role in the peri-implantation uterus. Further studies are needed to ascertain this immune suppression role of M2 MΦs in the peri-implantation uterus.

One surprising finding of this study is the expression of ly6C mRNA and protein in cells that are in contact with the implanted embryo following the removal of the uterine luminal epithelium. This Ly6C protein expression seems to be in decidual cells as we did not see the presence of CD45^+^ immune cells nearer to the implanted blastocyst in late day 5 and early day 6 of pregnancy. Previous studies have shown that the expression of Ly6C is not limited to hematopoietic cells. Ly6C protein expression has been described in intestinal and corneal epithelial cells [53,54]. The role of Ly6C expression in decidual cells is not understood. It is possible that Ly6C induces apoptosis of decidual cells as these cells are known to undergo cell death [55,56]. This proposition is made because some Ly6C^+^ cells are apoptosis-prone populations [57]. Ly6C^+^ immune cells are also known to perform both proinflammatory and immunosuppressive functions [58,59]. As the EBIS is considered a fine-tuned inflammatory as well as an immunosuppressive area, it is expected that Ly6C^+^ decidual cells may behave like local innate immune cells with pro-inflammatory and immunosuppressive roles supportive of embryonic survival. Although, we did not test whether these Ly6C^+^ decidual cells are, in fact, transitioned to immunomodulatory phenotype or not, requiring further investigation.

A variety of preclinical models use intraperitoneal LPS injection in mice to study infection-induced early and late pregnancy complications such as implantation defects, miscarriages and preterm birth. In this study, we observed that i.p. LPS, but not MPLA, a detoxified form of LPS, administration on day 5 of pregnancy caused dose-dependent defects at the EBIS by day 7 of pregnancy suggesting maternal infection-induced (inside-in) vulnerability of the EBIS. However, how systemic LPS induces defects at the EBIS remains unclear. We found that LPS-induced abnormal changes at the EBIS were coincident with a decrease in the number of MΦs and DCs from the myometrium and a huge infiltration of NPs into the NDSZ of the EBIS. The decreased number of MΦs and DCs in the myometrium may be due to an increased exit of these monocytes from the myometrium to the circulation in response to LPS suggesting that myometrial monocytes may not have a direct role in LPS-induced defects at the EBIS. Another possibility is that LPS triggers necrotic death of these myometrial monocytes leading to the release of cytokines that help to facilitate the migration of NPs to the inside of the EBIS by increasing the permeability of the local endothelial and myometrial cells [60]. The present study demonstrated that after 2 h of administering LPS, there was a marked transcriptional upregulation of two neutrophil chemoattractants *Cxcl1* and *Cxcl2* at the BIS. We interpret from these findings that LPS induces rapid transcription of neutrophil chemoattractant genes which in turn elicit neutrophil migration to the EBIS. It is conceivable that LPS-induced *Cxcl1* and *Cxcl2* occur in the EBIS resident cells, most of which are endometrial stromal cells. One striking observation made here is that only a few migrated neutrophils reach the DSZ; the majority remain within the NDSZ. It is possible that given the restricted permeability to macromolecules [61], the DSZ did not allow NP invasion closer to the implanted blastocyst even after LPS exposure. Multiple studies have demonstrated that while NPs are essential for host defense against infection, massive NP accumulation can also be detrimental to the surrounding tissue as seen in several diseases like lung injury, experimental colitis and rheumatoid arthritis [62]. Thus, LPS-induced NP recruitment within the NDSZ is likely a contributing component in the pathogenesis at the EBIS. NPs are known to release harmful substances, such as reactive oxygen species (ROS), tumor necrosis factor-alpha (TNFα), thromboxane, and matrix metalloproteinase-8 (MMP8), all of which could cause tissue dysfunction [63,64]. However, we observed no protection of LPS-induced early pregnancy defects following inhibition of NP recruitment via administration of anti-Ly6G suggesting that LPS-induced NP accumulation at the EBIS is not the primary cause for the observed EBIS defects. This finding is not surprising since an earlier study reported that even decidual NP infiltration in response to LPS during late gestation is not required for inducting preterm birth in the mouse model [65]. Thus, mechanisms other than NP infiltration contribute to LPS-induced effects at the EBIS.

Toll-like receptor 4 (Tlr4) is the major LPS signaling receptor since mice devoid of *Tlr4* are hyporesponsive to LPS [25]. We showed in one of our previous publications that Tlr4 and Cd14 genes are expressed within the decidua of the normal EBIS [28]. A recent in vitro study also showed inhibitory effects of LPS on decidual cell growth [66]. In this study, we provided evidence that EBISs respond to LPS by increased expression of mRNA of MyD88 and its downstream inflammatory cytokines genes including *Tnfα*, *Il1β* and *Il6*. Surprisingly, we did not detect any significant expression of Trif and its downstream gene *Ifnβ* by LPS. Together with our previous study in CD1 mice [28], these data in C57BL/6J also highlight that LPS can directly influence the BIS by engaging Tlr4-mediated MyD88, but not Trif, pathway. However, it remains largely unknown how LPS selectively activates the Tlr4-mediated MyD88 signaling pathway at the EBIS. Differentially expressed miRNAs in response to LPS challenge could provide a possible explanation for these results. miRNAs are small, single-stranded, noncoding RNA oligonucleotides that downregulate gene expression at the posttranscriptional level either by translational repression or mRNA degradation [67]. Studies have demonstrated changes in miRNA expression following LPS exposure in immune cells emphasizing their role in the immune response [67]. Since miRNAs are also expressed in uterine and embryonic cells [68], future studies are needed to identify LPS-induced global miRNA expression profile at the EBIS for better understanding the role of miRNAs in the regulation of the MyD88-dependent Tlr4 signaling pathway at this site.

A few studies have provided evidence that LPS when administered either systemic or intrauterine reaches the uterine tissue, placenta and fetus causing fetal injury [69,70], although this contention is contradicted by others [71]. Thus, we cannot exclude the possibility that LPS-induced systemic inflammation may yet be a risk factor for inducing the defects at the EBIS. The intriguing history of acute phage response induced by systemic administration of LPS suggests that this endotoxin induces immediate production and release of proinflammatory cytokines including Tnf, Il1β or Il6 in serum/plasma from various types of leukocytes such as monocytes, polymorphonuclear leukocytes and lymphocytes and endothelial cells [66,72,73]. Thus, it is possible that increased levels of systemic inflammatory mediators which by reaching the decidua move their concentration limits to adverse-effect levels within the decidua. In this context, there is evidence that (1) elevated circulating Il-1β and Tnf in first trimester pregnancies reflected a negative obstetrics outcome [74]; and (2) premature labor due to maternal infection is mediated by Il-1β and Tnf-α [75]. The EBIS has also been identified as a source of cytokine production and the locally produced cytokines are known to play a key role in maintaining homeostasis at the EBIS [76]. Thus, when the levels of cytokines are too high at the BIS due to the arrival of infection-induced cytokines from the circulation and/or production of local cytokines, healthy cells at the EBIS are ruined resulting in defects. At this stage, we favor the idea that LPS-induced defects at the EBIS may result from both direct and indirect actions of LPS. However, a follow-up study is needed to establish between direct and indirect actions of LPS at the EBIS which one is eventually accountable for inducing defects at the EBIS.

## 4. Materials and Methods

### 4.1. Animal and Tissue Preparation

All experiments in mice were conducted with the approval (Protocol# M1800011-00 and M1800011-01) of the Institutional Animal Care and Use Committee at Vanderbilt University, Nashville, TN. Virgin C57BL/6J male and female mice (7–8 weeks) were purchased from Jackson Laboratory, Bar Harbor, ME, USA. All studies were carried out in compliance with the ARRIVE guidelines [77]. Pregnancy was induced by caging three females with a fertile male overnight. Plug-positive mice in the following morning were designated as day 1 of pregnancy. Whole uterine horns were collected at ~21 h of day 4 of pregnancy. On day 5, EBISs were identified by distinct blue band formation along the uterus after an intravenous injection of 100 µL of Chicago Blue B solution in saline (1%) and harvested at ~09-h and/or ~21 h. EBISs on day 6 of pregnancy were distinct, visually identified and harvested at ~09 h. Harvested tissues were frozen immediately and stored at −80 °C.

### 4.2. LPS-Induced Inflammation at the EBIS and Early Pregnancy Defect

To verify the consequence of intraperitoneally (i.p.) instilled LPS on early pregnancy in C57BL/J mice, we first titrated LPS dosing for early pregnancy complications. LPS [*E. coli* (serotype 055: B5; Catalog# L2880) from Sigma-Aldrich, St. Louis, MO, USA] was dissolved in sterile saline and sonicated before use. Pregnant mice received single injection (i.p.) of four doses of LPS (0.1, 0.5, 0.75 or 1 µg) in 100 µL saline on the morning (~09 h) of day 5 of pregnancy. Animals in the control groups were injected (i.p.) with either equivalent volume of vehicle (sterile saline) or the highest dose (1 µg) of MPLA (Catalog# L6895, Sigma-Aldrich), a detoxified form of LPS, that does not trigger inflammatory response [28]. Animals from all groups were euthanized on ~48 h post-treatment (~day 7 of pregnancy) to observe the effect of LPS on pregnancy outcome. EBISs from each mouse were visualized, counted, weighed and quickly frozen. A group of animals from saline-,1 µg LPS- and 1 µg MPLA-treated groups were euthanized 2- or 24-h post-injection (day 6) and EBISs were harvested for studying inflammatory response at the EBIS.

### 4.3. Depletion of NPs Prior to LPS Injection in the Pregnant Mice

Previous studies have demonstrated that an i.p. injection of anti-ly6G antibody in mice resulted in successful depletion of circulating neutrophils as examined by flow cytometry [78,79,80]. Thus, to transiently deplete NPs in vivo, a group of mice was injected (i.p.) with 100 µL of solution containing 150 µg of anti-ly6G (Catalog# BP0075-1; BioXCell, Lebanon, NH, USA) antibodies on day 4 of pregnancy 24 h prior to receiving 1 µg LPS on day 5 of pregnancy. Similarly, a group of control mice received an injection (i.p.) of IgG2b isotype antibody (Catalog# BE0089; BioXCell) at a dose of 150 µg in 100 µL of solution. Animals were euthanized either on day 6 to confirm NP depletion from the EBIS by immunofluorescence study or day 7 of pregnancy to ascertain pregnancy outcome as described in the above section.

### 4.4. Immunofluorescence

The antibodies used in this study with their sources, dilutions and targets are listed in Appendix A. Total leukocyte population was first identified by detection of CD45 protein [31]. Next, each subtype of leukocytes was identified using their markers. Antibody against Ly6G, a GPI anchored cell surface protein [34] antigens was used for NPs. Total monocyte population was identified using antibodies against ly6C, a GPI anchored cell surface protein antigen. Monocytes are precursors of MΦs and DCs. MΦs were identified using antibodies against F4/80/Ly71 [36] and CD206 [37] antigens. DCs were identified using antibodies against CD11c/Integrin (a membrane protein) antigen [38]. Immunofluorescent detection method instead of flow cytometry was used here as this is an effective tool to visualize the normal as well as abnormal location and distribution of leukocytes in separate compartments of the uterus and EBIS in various situations.

Normally two to three cryosections (12 µm) from each uterine tissue were laid on Superfrost^TM^ plus Microscope Slides and stored at −80 °C until use. On the day of experiment, sections were thawed and fixed for 10 min in stated cold fixative for each antibody in Appendix A. The slides were washed with phosphate-buffered saline (PBS). Non-specific binding of antibody in tissue sections was blocked using 10% goat serum (Catalog# 500627; Thermo Fisher Scientific, Waltham, MA, USA) in which the second antibody was raised for 30 min. After discarding the excess liquid from slides the primary antibody was added as detailed in Appendix A. One slide was used for a mock staining without the addition of the primary antibody. After an overnight incubation at 4 °C slides were thoroughly washed and a suitable secondary antibody with 4′,6-diamidino-2-phenylindole (DAPI; Catalog# D9542, Sigma Aldrich) was applied for 2 h at room temperature. Slides were washed with PBS and cover slipped with Fluoromount G (Catalog# 0100-01; SouthernBiotech, Birmingham, AL, USA). The sections were left for drying in the dark before visualization and imaging under a Nikon Eclipse TE2000E inverted microscope equipped with X-Cite 120 for fluorescence and Opti-grid structured light confocal imaging system (Boyce Scientific, Inc., Gray Summit, MO, USA). The image of each section was captured in various magnifications. As immune cells are usually found in the intestinal tissue, this tissue was chosen to test specificities of primary and secondary antibodies used in this study prior to their application in the uterine tissue. The specificities of antibodies were validated with and without primary antibody controls, and these data were presented in Appendix A.

To quantitate the number of leukocytes and their subtypes present in separate compartments of EBIS, images of sections were captured at 40x. Each compartment of the EBIS such as the myometrium, NDSZ or DSZ was manually annotated using ImageJ 1.45s software (National Institutes of Health, Bathesda, MD, USA). Immunopositive cells were manually counted within each compartment avoiding bias and expressed as number or fold change per mm^2^.

### 4.5. Quantitative RT-PCR

Total RNA was isolated from pooled EBISs from each mouse using TRIzol reagent (Catalog# 15596026, Life Technologies, Carlsbad, CA, USA). The quantity of all preparations was determined by NanoDrop 2000 spectrophotometry (Thermo Fisher Scientific). About 1 µg of total RNAs after removal of genomic. First-strand cDNA was synthesized using SuperScript™ VILO™ cDNA Synthesis Kit (Catalog# 11754-050, Invitrogen by Thermo Fisher Scientific). RT-PCR was performed using a CFX96^TM^ Real-Time System (Bio-Rad, Hercules, CA, USA) and PowerUp SYBR Green Master Mix (Catalog# A25742, Applied Biosystems by Thermo Fisher Scientific). The conditions used for qPCR were as follows: 95 °C for 2 min followed by 39 cycles of 95 °C for 15 s, 55 °C for 30 s and 60 °C for 1 min. All reactions were run in triplicate. The primers used for qPCR are listed in Appendix A. The 2^−∆∆Ct^ method was used to quantify relative gene expression using the glyceraldehyde 3-phosphate dehydrogenase (GAPDH) housekeeping gene as an internal control [81] and expressed as fold change relative to expression in the vehicle group. The values reported are the mean (±SD) of three replicates.

### 4.6. In Situ Localization of Ly6C mRNA

RNAscope in situ hybridization (Advanced Cell Diagnostics (ACD), Biotechne, Newark, GA, USA) was performed in frozen sections according to the manufacturer’s instructions. Briefly, frozen sections were fixed 10% formalin solution (Catalog# SF98-4; Thermo Fisher Scientific), washed in PBS, dehydrated in graded alcohols (50%, 70% and 100%), dried in air before addition of hydrogen peroxide, washed in water, digested in protease IV (Catalog# 322336, ACD), and washed in PBS followed by a series of hybridization with a mouse *Ly6C1* probe (Catalog# 558121, ACD) and reagents from Multiplex Fluorescence detection kit V2 (Catalog# 322350, ACD). Following hybridization, slides were washed and fluorescein (Catalog# NEL 741E001KT; Perkin Elmer, Waltham, MA, USA) was added to slides. Sections were cover slipped with Prolong Gold Antifade Reagent (Catalog# P36934; Invitrogen by Thermo Fisher Scientific). Probe binding was visualized as described in the immunofluorescence methodology section. Positive and negative control experiments were also performed to assess the specificity of our RNAscope. Sections were stained with mouse *Alpl* (Catalog# 441161, ACD) and *Dapb* (Catalog# 320871, ACD) probes as positive and negative controls, respectively. Alpl gene is known to express in decidual cells of the day 6 EBIS [82]. Dapb probe was used as a negative ISH control because this probe is against a bacterial (*Bacillus subtilis)* gene that is not present in mouse tissue.

### 4.7. Statistics

Data were presented as means ±SEM. Statistical analysis was performed using a one-way analysis of variance followed by Tukey’s post hoc test. All statistical analyses were performed using GraphPad Prism 5.0 software (GraphPad, San Diego, CA, USA). A *p*-value <0.05 was considered statistically significant.

## 5. Conclusions

From this study, we conclude the following: (1) migration of endometrial leukocytes out of the core EBIS may be a normal process fundamental to the blastocyst implantation physiology and relevant to immune evasion by the blastocyst; (2) although inflammatory LPS exposure to the mother causes overwhelming migration of circulating NPs to the endometrial region outside the core EBIS and defects at the EBIS, these migrated NPs are not involved in inducing pathophysiology at EBISs; and (3) given that LPS induces transcription of MyD88 and MyD88-dependent proinflammatory cytokines genes like *Tnfα*, *ll6*, and Il1β but not Trif and Trif-mediated Ifnβ, signaling through the MyD88-dependent pathway may be the driver of LPS-induced defects at the EBIS. These observations provide a mechanism by which Gram-negative bacteria can precipitate EBIS changes associated with early pregnancy loss. Further studies are undergoing to confirm the relevance of the MyD88-dependent LPS/Tlr4 pathway in inducing LPS-induced defects at the EBIS such that therapies targeted at blocking LPS signaling for the management of infection-induced early pregnancy disorders can be found.

## Figures and Tables

**Figure 1 ijms-22-07932-f001:**
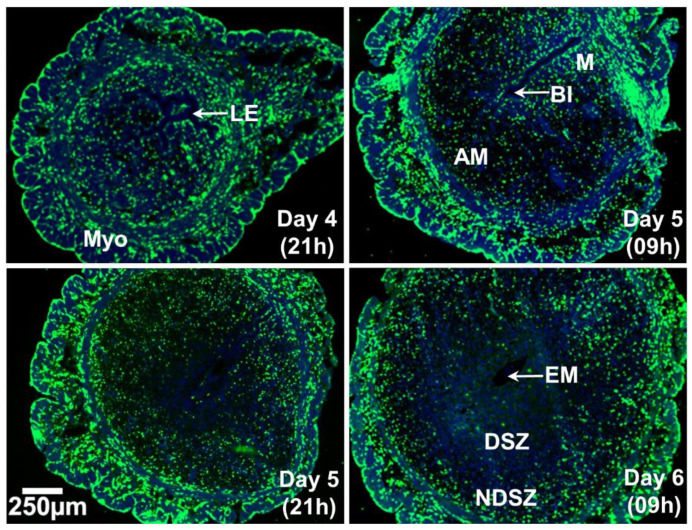
Event-specific change in the distribution pattern of uterine CD45^+^ immune cells at 12 h intervals during the peri-implantation period (days 4-6) of pregnancy. Cross sections obtained from two pieces of uterine tissues/mouse/timepoint of pregnancy were fixed, immunostained with anti-CD45 (green), counterstained with DAPI (blue) and photographed. At least three pregnant mice were used at each timepoint of pregnancy. AM, anti-mesometrial side; BL, blastocyst; DSZ, decidualized stromal zone; EM, embryo; LE, luminal epithelium; M, mesometrial side, Myo, myometrium; NDSZ, non-decidualized stromal zone.

**Figure 2 ijms-22-07932-f002:**
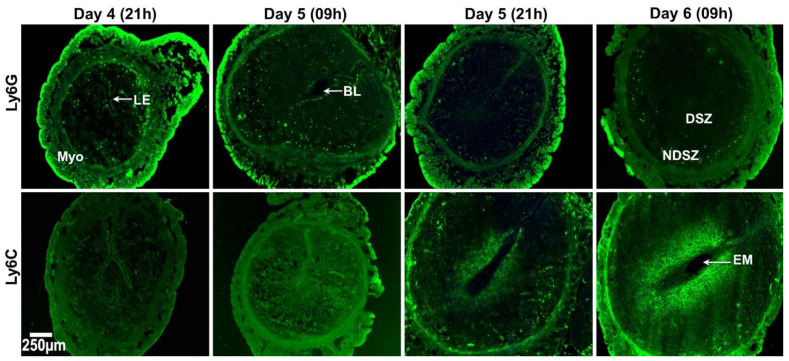
Event-specific change in the distribution pattern of uterine neutrophils and monocytes in the receptive uterus of day 4 (09h) and EBISs of days 5 (09h and 21h) and 6 (09h) of pregnancy. Cross sections obtained from two pieces of uterine tissues/mouse/timepoint of pregnancy were fixed, immunostained with anti-Ly6G and anti-Ly6C antibodies (green), counterstained with DAPI (blue) and photographed. At least three pregnant mice were used at each timepoint of pregnancy. BL, blastocyst; DSZ, decidualized stromal zone; EM, embryo; LE, luminal epithelium; Myo, myometrium; NDSZ, non-decidual stromal zone.

**Figure 3 ijms-22-07932-f003:**
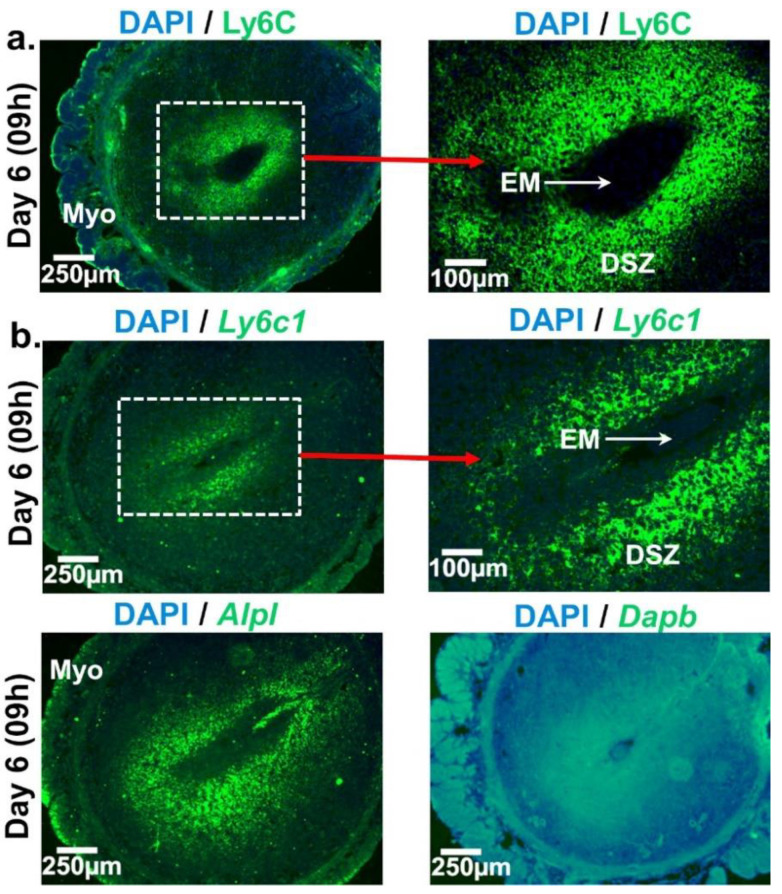
Cell-specific localization of Ly6C protein and mRNA at the day 6 EBIS. (**a**) Ly6C protein expression by immunofluorescence and (**b**) Ly6C mRNA expression by in situ *hybridization* (upper panel) and images of positive and negative controls for RNAscope (lower panel). Alpl and Dapb mRNA expressions were examined as positive and negative controls, respectively. Cross sections obtained from two blastocyst implantation sites/mouse/timepoint of pregnancy were fixed, immunostained with anti-Ly6C (green), counterstained with DAPI (blue) and photographed. The Ly6C protein and mRNA expressing areas were shown in higher magnification in right side images of Figure 3a,b, respectively. At least three pregnant mice were used at each timepoint of pregnancy. DSZ, decidualized stromal zone; EM, embryo; Myo, myometrium.

**Figure 4 ijms-22-07932-f004:**
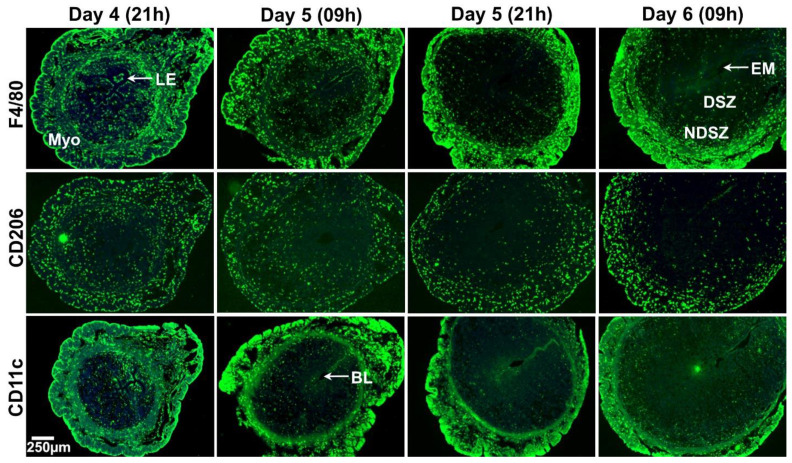
Event-specific change in the distribution pattern of uterine MΦs and DCs during the peri-implantation period of pregnancy. Immunofluorescence detection of F4/80, CD206 and CD11c positive cells. Cross sections obtained from two pieces of uterine tissues/mouse/timepoint of pregnancy were fixed, immunostained with F4/80, CD206 and CD11c (green), counterstained with DAPI (blue) and photographed. At least three pregnant mice were used at each timepoint of pregnancy. BL, blastocyst; DSZ, decidualized stromal zone; EM, embryo; LE, luminal epithelium; Myo, myometrium; NDSZ, non-decidualized stromal zone.

**Figure 5 ijms-22-07932-f005:**
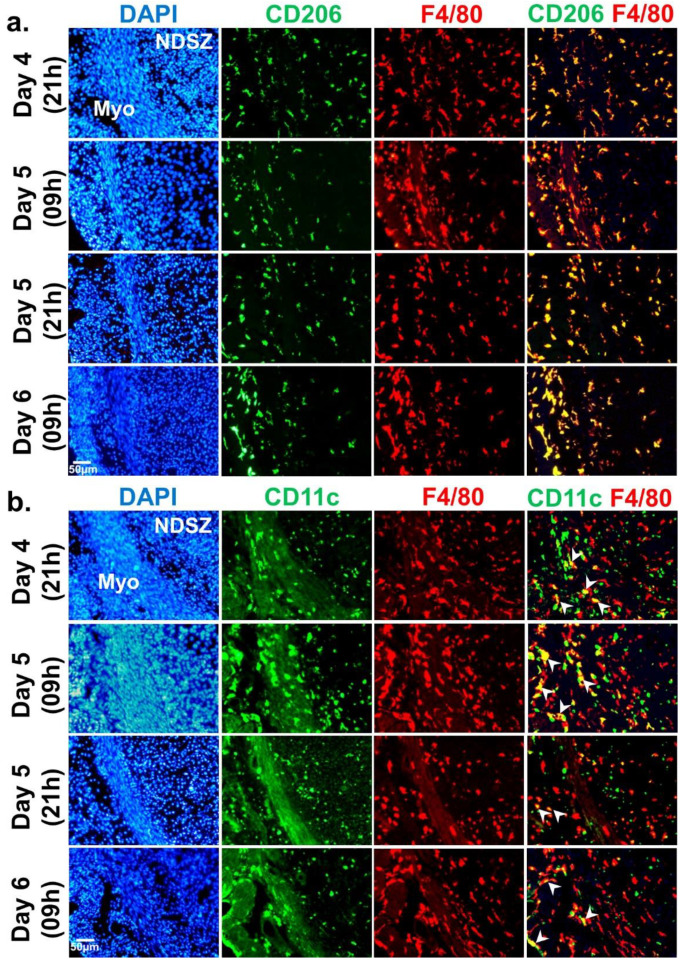
Identification of M1 and M2 MΦs in the uterus during the peri-implantation period (days 4–6) of pregnancy. (**a**) Immunofluorescence colocalization analysis of F4/80 (red) and CD206 (green) proteins in the receptive uterus of day 4 (09 h) and EBISs of days 5 (09 h and 21 h) and 6 (09 h) of pregnancy. Cross sections obtained from two pieces of uterine tissues/mouse/timepoint of pregnancy were fixed, immunostained, counterstained with DAPI (blue) and photographed. Yellow colored spots designated the colocalization of F4/80 and CD206 proteins. In order to show the tissue structure, overblown DAPI stained images are presented in left-hand column; (**b**) Immunofluorescent double staining was performed using anti-F4/80 (red) and anti-CD11c (green) antibodies. Cross sections obtained from two pieces of uterine tissues/mouse/timepoint of pregnancy were fixed, immunostained with anti-F4/80 and anti-CD11c antibody, counterstained with DAPI (blue) and photographed. Arrowheads designated the colocalization of F4/80 and CD11c proteins. All images were taken from the anti-mesometrial side of the uterus. At least three pregnant mice were used at each timepoint of pregnancy. Myo, myometrium; NDSZ, non-decidualized stromal zone.

**Figure 6 ijms-22-07932-f006:**
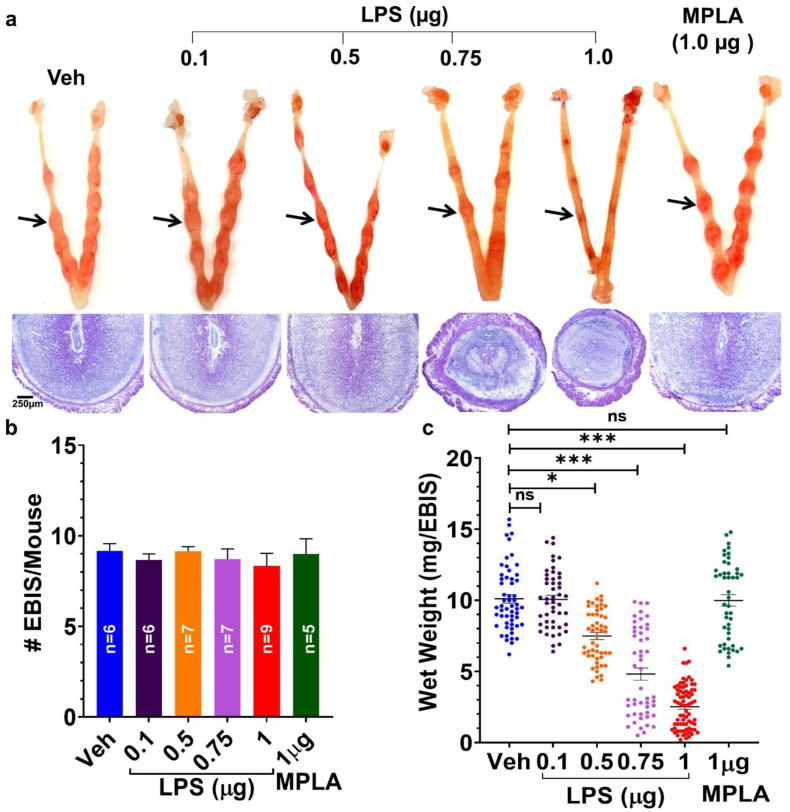
Systemic lipopolysaccharide (LPS) exposure induced dose-dependent defects at the EBIS. All pregnant mice except the vehicle (saline) and monophosphoryl lipid A (MPLA) controls received an injection (i.p.) of LPS on day 5 of pregnancy and their pregnancy outcomes were analyzed on day 7. (**a**) Visible structural appearances of EBISs (top row) and their histopathology (H&E staining, bottom row) determined the dose-dependent effects of LPS exposure on EBISs. Arrows indicate blastocyst implantation sites; (**b**) Number of EBISs (number (n) of mice used in each treatment group was indicated within the bar); (**c**) Wet weight of individual EBIS. Results are presented as mean ±SEM. * *p* < 0.05, *** *p* < 0.001 (ANOVA followed by Tukey test). Veh, vehicle; EBIS, early blastocyst implantation site; ns, not significant.

**Figure 7 ijms-22-07932-f007:**
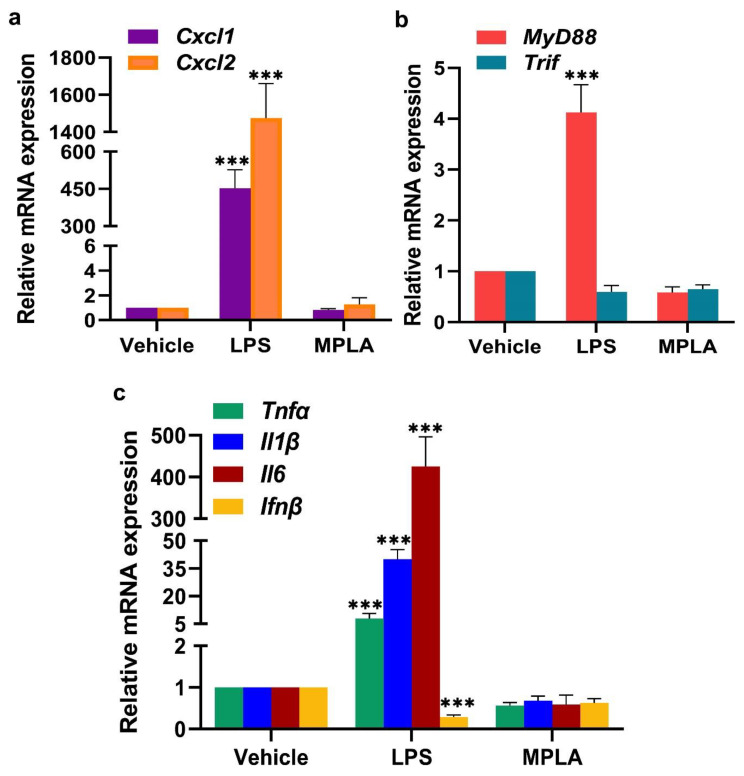
Maternal LPS exposure elevated expression of mRNAs of chemokines and MyD88 and MyD88-modulated cytokines, but not Trif and Trif-modulated cytokine at the EBISs. Expressions of mRNAs of chemokines, Cxcl1 and Cxcl2 (**a**), adaptor proteins, MyD88 and Trif (**b**) and cytokines, TNFα, Il1β, Il6, and Ifnβ (**c**) were determined by quantitative RT-PCR. All EBISs were collected 2 h after vehicle or MPLA (1 µg) or LPS (1 µg) injection (i.p.). Data were normalized to GAPDH, expressed as Mean ± SD of three experiments and one-way ANOVA followed by Tukey’s test was applied for statistical analysis. *** *p* ≤ 0.001.

**Figure 8 ijms-22-07932-f008:**
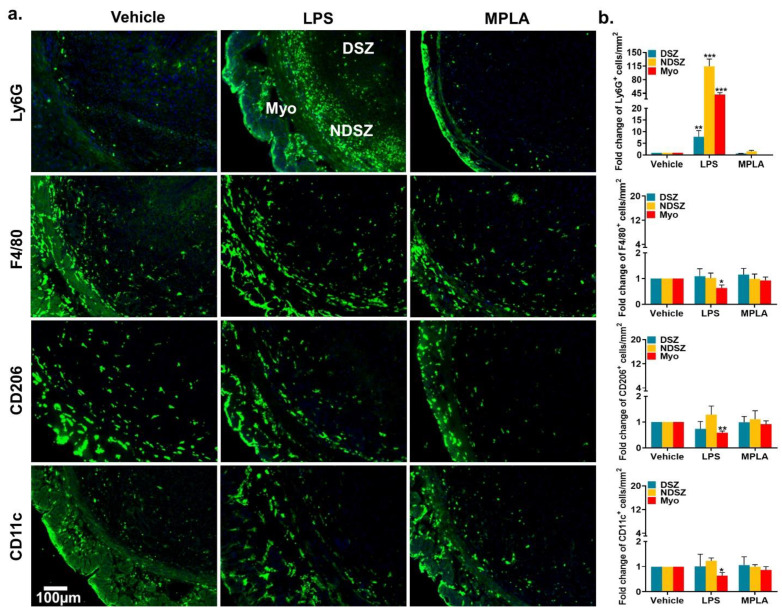
Maternal LPS exposure triggered only neutrophil infiltration at the EBIS. Vehicle (saline), MPLA and LPS (1µg/mouse) were intraperitoneally injected on day 5 of pregnancy and neutrophils, macrophages and dendritic cells were identified at the EBIS 24 h later. (**a**) Representative immunofluorescence images of EBISs from vehicle, LPS and MPLA treatment groups. Here neutrophils were stained with Ly6G antibody, macrophages were identified using F4/80 and CD206 antibodies, DCs were stained with CD11c antibody. Cross sections obtained from two pieces of uterine tissues/mouse/dose of day 6 of EBIS were fixed, immunostained with respective primary antibodies, counterstained with DAPI (blue) and photographed. At least six pregnant mice were used for each treatment. DSZ, decidualized stromal zone; Myo, myometrium; NDSZ, non-decidualized stromal zone; (**b**) Quantification of immune cells at the EBIS following vehicle, LPS and MPLA treatment. Total number of Ly6G^+^, F4/80^+^, CD206^+^ and CD11c^+^ cells were counted in three distinct regions (DSZ, NDSZ and Myo) of the EBIS. At least three immunostained sections from each treatment group were used for immune cell counting. Results are shown as the fold change relative to vehicle control and represent the mean ± SD. * *p* < 0.05, ** *p* < 0.01, *** *p* < 0.001 (ANOVA followed by Tukey test).

**Figure 9 ijms-22-07932-f009:**
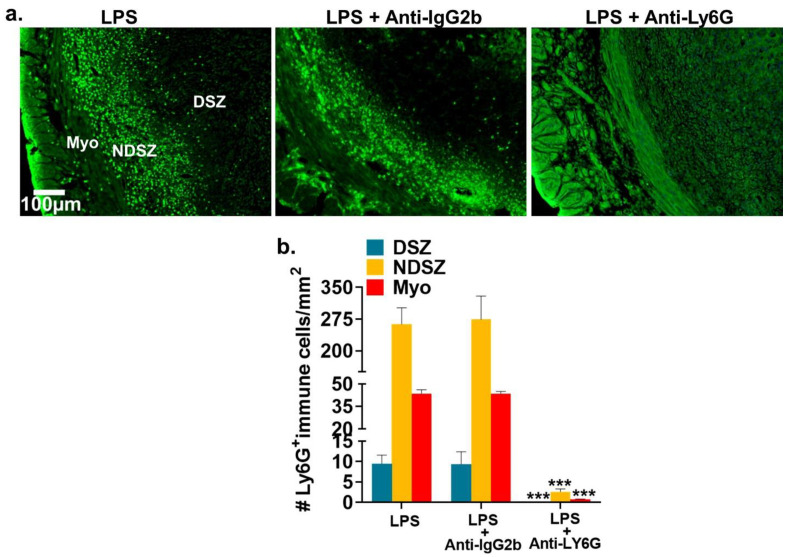
LPS-induced neutrophil recruitment at the EBIS (Day 6) is blocked by prior anti-Ly6G antibody treatment. Anti-Ly6G and anti-IgG2b antibodies (150 µg/mouse) were injected (i.p.) on day 4 of pregnancy prior to LPS (1µg/mouse) injection on day 5 of pregnancy. BISs were excised 24 h after LPS (day 6 of pregnancy). Neutrophils were identified in cross sections of the BIS using NP-specific anti-Ly6G antibodies. At least five pregnant mice were used for each treatment. Cross sections obtained from two pieces of uterine tissues/mouse/dose of day 6 of BIS were fixed, immunostained with anti- Ly6G (green), counterstained with DAPI (blue), photographed and counted. (**a**) Representative immunofluorescence images of EBISs from LPS, LPS plus anti-IgG2b and LPS plus anti-Ly6G treatment groups. Neutrophils were stained with Ly6G antibody. DSZ, decidualized stromal zone; Myo, myometrium; NDSZ, non-decidualized stromal zone; (**b**) Quantification of immune cells at the EBIS from LPS, LPS plus anti-IgG2b and LPS plus anti-Ly6G treatment groups. Total number of Ly6G^+^ cells were counted in three compartments (DSZ, NDSZ and Myo) of the EBIS. At least three immunostained sections from each treatment group were used for immune cell counting. Results are shown as number of Ly6G cells per mm^2^ area and represent the mean ±SD. *** *p* < 0.001 (ANOVA followed by Tukey test).

**Table 1 ijms-22-07932-t001:** Neutrophil depletion failed to protect LPS-induced implantation site defects. **#** number; *** *p* < 0.005, Significatly different from group 1.

Group	Treatment	Mice (#)	EBIS/Mouse (#)	Wet Weight/EBIS (mg)
1	Saline (vehicle)	11	6.7	14.73 ± 0.18
2	Anti-Ly6G Ab	5	6.4	14.64 ± 0.30
3	LPS	13	6.6	3.43 ± 0.24 ***
4	Anti-IgG2b + LPS	5	6.2	3.69 ± 0.28 ***
5	Anti-Ly6G Ab + LPS	5	6.2	3.92 ± 0.29 ***

## Data Availability

The datasets generated and/or analyzed for the current study are available from the corresponding author on reasonable requests.

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
