# Peer review of "Maternal Neutrophil Depletion Fails to Avert Systemic Lipopolysaccharide-Induced Early Pregnancy Defects in Mice"

_ijms, 2021, doi:10.3390/ijms22157932_

Round 1
Reviewer 1 Report
The manuscript by Panja et al. describes a study of innate immune cell populations in the periimplantation mouse uterus and the possible mechanisms of systemic LPS-induced early pregnancy defects. The paper is well written and clearly structured. The results are presented to the point and appropriately discussed. This Reviewer has only minor comments.
The title does not really represent the total content of the study, in fact, only half of the investigations.
Introduction: When discussing the role of different types of immune cells in the process of implantation, more recent reviews could be cited. I believe that many more details about types and subpopulations of immune cells are known than in the review by Hunt from the year 1994.
Lines 30-32: Adding these numbers suggests almost 100% pregnancy failure.
Line 57: What does “exit” mean?
Line 55-57: Reference is missing for this sentence.
Line 65-67: The Introduction should mainly contain the state-of-knowledge. The hypotheses of the authors should be at the end of the Introduction.
Lines 67-78: I am not sure that all relevant papers about regulation of the innate immune response are mentioned. For example, there is one paper by Tirado-González et al. PLoS One 2012 describing the role of NK cells in regulating dendritic cells and decidual development. This paper supports also the statement in lines 75-77.
Line 158: Here, the authors are talking about mRNA in Fig. 3b. In the figure legend, it is written “gene” and “gene expression”. This should be changed to mRNA.
Line 220: Write “mRNA levels” here or “mRNAs” instead of “genes” in line 221.
Line 385: Should this be “pre-implantation”? In line 386 it is stated that Ly6C+ cells are present near the implanted blastocyst.
Line 429: word missing after “embryo”
Line 601: sentence incomplete
Author Response
Reviewer 1: We are grateful for constructive comments of this reviewer.
1) About the title: We acknowledge this suggestion. However, we have decided to keep the title as it is since the current title reflects the gripping and most meaningful aspects of the study. Below are the reasons for our decision.
In the first part of this manuscript, we have demonstrated a comprehensive picture of changes in the distribution patterns of leukocytes at the uterine blastocyst implantation site as the early pregnancy advances from day 5 to day 6. This temporal profile provides us information about how blastocyst implantation affects local uterine immune cell movement and distributions. This information served as a foundation for pursuing a mechanistic approach to define the role of neutrophils at the implantation site in response to an inflammatory insult. The latter part demonstrated that while LPS causes massive neutrophil infiltration at the implantation site and implantation site defects yet blocking of infiltration has no influence on LPS-induced pregnancy defects. Thus, we feel that this mechanistic part of the study is better suited for the title of the manuscript as it will attract the attention of the reader at first glance.
2) Citation of recent papers regarding the role of immune cells: We have now acknowledged this and incorporated several recent studies in the introduction section.
3) Lines 30-32: Adding these numbers-------pregnancy failures.
We have rephrased this sentence to avoid possible misunderstanding.
4) Line 57: What does exit mean?
This is a typing error. It is now corrected in the text.
5) Reference is missing for this sentence.
Reference is provided.
6) Lines 65-67. The introduction should-----------------at the end of the introduction.
This sentence has been modified
7) Lines 67-78: I am not sure that all relevant papers------------------------the statement in lines 75-77.
We have incorporated a few more relevant papers.
8) Line 158: Here, the authors are--------------------changed to mRNAs.
Changes have been made
9) Line 220: Write mRNA levels---------------instead of genes in line 221.
These changes have been incorporated.
10) Line 385: Should this be-----------near the implanted blastocyst.
We have decided not to make any changes here since we are talking about Ly6C positive monocyte in the first sentence and Ly6 positive decidual cells in the following sentence.
11) Line 420: Word missing after the embryo.
We have added the word “implantation”.
12) Sentence incomplete
We have corrected this mistake.
Reviewer 2 Report
The authors could demonstrate straightforward that the blocking activation of Tlr4-MyD88 signaling pathway could be one of the effective approach to prevent infection-induced pathology at EBISs.
Author Response
Reviewer 2:
We acknowledge this comment. Your point will be the focus of our next study. Currently, we are assessing the anti-inflammatory properties of several TLR4 antagonists belonging to the different classes of compounds in correcting LPS-induced proinflammatory gene expressions in the uterus as well as in uterine stromal cells in culture. These studies are important as TLR4 antagonists have diverse mode of action in blocking the TLR4 signaling. After completion of these studies, the most promising and potent TLR4 antagonist will be used to prevent LPS-induced blastocyst implantation site defects. Thank you for the suggestion.
Reviewer 3 Report
I read with great interest the manuscript, which falls within the aim of this Journal. In my honest opinion, the topic is interesting enough to attract the readers’ attention. Nevertheless, authors should clarify some points and improve the discussion, as suggested below.
Authors should consider the following recommendations:
- Manuscript should be further revised in order to correct some typos and improve style.
- Recent and novel evidence suggested that epigenetic changes, in particular altered expression of selective miRNA, may play a key role in early gestational complications. It would be mandatory to discuss (at least briefly) this topic.
Author Response
Reviewer 3:
1) Manuscript should be further revised------------------some typos and improve style.
Manuscript have been thoroughly revised
2) Recent and novel evidence suggested that---------------------to discuss (at least briefly) this topic.
According to your suggestion we have now introduced a few sentences about the possible role of miRNAs in activation of MyD88-dependent Tlr4 signaling pathway at the blastocyst implantation site following maternal LPS exposure in the discussion section of the manuscript.